# A ZKP-based anonymous biometric authentication scheme for the E-health systems

**Xuechun Mao[1], Xiaqing Zhou[2], Xiaoming Zhao[1], Ying Chen[1]***

**1** Department of Computer Science, Taizhou University, Taizhou, Zhejiang, China, **2** Medical Records Division, Taizhou Hospital, Taizhou, Zhejiang, China

* ychen222@tzc.edu.cn

**Data availability statement:** All relevant data are within the paper and its Supporting Information files.

## Abstract

The widespread adoption of e-health systems raises critical concerns regarding data privacy and network security. Ensuring secure and reliable data sharing between patients and healthcare professionals remains a significant challenge. To address this, we propose a novel anonymous authentication scheme tailored for e-health environments, integrating zero-knowledge proof (ZKP) with multimodal biometrics. Our key contributions are as follows: (1) applying the Pedersen vector commitment algorithm to construct a biometric-based ZKP scheme, thereby ensuring enhanced security and privacy-preserving authentication; (2) utilizing multimodal cancelable biometrics generate (MCBG) technology, integrating fingerprint, face, and iris modalities to strengthen the security of the verification process; and (3) providing a detailed security analysis that demonstrates our scheme meets essential security requirements, including anonymity, authenticity, unlinkability, forward security, and resistance to replay attacks. Experimental results demonstrate stable proving and verification time of approximately 78 ms and 140 ms, respectively, regardless of the proof range, validating its efficiency and practicality for secure authentication in e-health systems.

## Introduction

The advent of innovative technologies, notably mobile cloud computing (MCC) and the Internet of Medical Things (IoMT), has sparked profound transformations within the healthcare industry, particularly in e-health [1,2]. These advancements facilitate the sharing of electronic health records (EHRs) among different healthcare entities, which reduces the risk of errors, duplication of tests, and delays in treatment. When a patient visits a hospital or medical institution, his EHRs are uploaded to the cloud server. Subsequently, healthcare professionals can access these records after obtaining the patient's authorization. Simultaneously, patients can conveniently view their medical history, test results, and treatment plans online instead of relying on paper records. However, the growing EHRs stored in e-health systems present significant security challenges [3–5]. For example, unauthorized entities may gain malicious access to EHRs without permission, which harms the data integrity, privacy, and security of

**Funding:** This research is partially supported by the Project under the Soft Science Research Project of Zhejiang Province of China (2025C35030) and the Key Program of the Natural Science Foundation of Zhejiang province of China (No. LZ20F020002). The funders had no role in study design, data collection and analysis, decision to publish, or preparation of the manuscript.

e-health systems. Therefore, it is necessary to propose efficient access control solutions for e-health systems.

Biometric authentication technology is an essential security mechanism in e-health systems, providing robust protection against identity forgery and unauthorized access. However, uni-modal biometric technologies often face challenges, low recognition accuracy, vulnerability to spoof attacks, and insufficient distinctiveness. To overcome these limitations, multimodal biometric technology[6] integrates features from two or more modalities, enhancing recognition accuracy and increasing security defenses against intrusions. Nonetheless, both uni-modal and multimodal biometric technologies exhibit inherent limitations that need to be addressed.

Biometric authentication technologies above are structured around two primary phases: enrollment and authentication. During the enrollment phase, reference templates are stored on servers under the protection of service providers. However, this process can lead to various security risks. Malicious service providers might misuse these templates, leading to unauthorized applications. Furthermore, security breaches may result in the leakage of these templates, thereby compromising user privacy. In the authentication phase, the system needs reference templates retrieved in plaintext to compute the Hamming distance and subsequent comparisons. This process significantly heightens privacy and security concerns, exposing sensitive biometric information to potential threats and misuse [5].

Consequently, numerous researchers have proposed biometric authentication schemes [7–10] to enhance security in e-health systems. Among them, the zero-knowledge proof (ZKP) protocol [11] enables a prover to convince a verifier that some statement is true without revealing anything more than the truth. In e-health systems, ZKP-based biometric authentication schemes can provide efficient and practical solutions without requiring complex security algorithms. However, many existing ZKP schemes [12] fail to comprehensively protect the entire authentication process, often leaving gaps in security and privacy or sacrificing efficiency. Our proposed scheme addresses these limitations by integrating multimodal cancelable biometrics with the Pedersen vector commitment algorithm, significantly enhancing authentication security and user privacy.

**Our contributions.** In order to solve the problems above, we propose a **Z**KP-based **A**nonymous **B**iometric **A**uthentication (ZABA) scheme utilizing multimodal cancelable biometrics generate (MCBG) technology. The main contributions are as follows:

- We present a ZABA scheme based on the combination of ZKP and elliptic curve cryptography (ECC). The proposed scheme effectively ensures patients' anonymity throughout the authentication process. Furthermore, our scheme satisfies the anonymity, authenticity, unlinkability, forward security, and replay-attack resistance, thereby preventing attackers from deducing any sensitive information about patients' identities during the authentication process.
- Combining MCBG technology with the biometric ZKP scheme, we establish a robust framework for secure authentication at the anonymous authentication level. This framework uses the patients' biometric data, including their fingerprint, face, and iris, to enhance their security in verifying their identity.
- We have implemented the ZABA scheme within the e-health system, and its effectiveness has been assessed through comprehensive evaluations. Compared to existing methods, ZABA leverages fingerprint, face, and iris biometrics to strengthen security, maintains stable computational performance (approximately 78 ms proving time and 140 ms verification time).

## Related work

**Anonymous authentication scheme.** Numerous researchers [13–15] have proposed anonymous authentication schemes to address the critical concern of user privacy. These schemes adopt different cryptographic techniques, including symmetric encryption, public key cryptography, ring signature, and ZKP, to ensure user anonymity throughout the authentication process. Arfaoui et al. [16] proposed a context-aware anonymous authentication and key agreement scheme to enhance selective anonymous authentication in healthcare systems. In 2023, Zhu et al. [17] proposed a lightweight anonymous authentication scheme that leveraged pseudonym techniques and symmetric encryption to improve the efficiency of anonymous device retrieval by the server. However, both schemes required an interactive key agreement process, which could be impractical in real-time authentication scenarios. Chinnasamy et al. [2] proposed an access control system that utilized blockchain technology coupled with the distributed storage of interplanetary file systems. This system resolved the secure data sharing within mobility computing environments. Zhang et al. [18] proposed an anonymous authentication scheme that combined group signatures with Merkle hash tree mechanisms, enabling multiple accesses after a single authentication. However, this scheme operated under a server- and gateway-centric architecture, where the gateway and its security was contingent on the robustness of the server and gateway security mechanisms.

Zero-knowledge authentication [12,19] is one of several useful techniques employed to ensure anonymous authentication. Current anonymous authentication schemes based on ZKP [20] are commonly regarded as efficient solutions without complicating security algorithms. Xi et al. [19] introduced an anonymous authentication scheme for the Internet of Vehicles (IoV) that combined ZKP with the Fujisaki-Okamoto commitment algorithm, enhancing authenticity. But, a notable limitation of this scheme was its traceability, which required a third-party trusted organization to implement. Zhang et al. [14] introduced an attribute-based multi-authority ciphertext-policy encryption scheme to construct an access control model suitable for the distributed Internet of Things (IoT) system. Within their authentication scheme, the user's information remained safeguarded against exposure to the server through utilizing ZKP. In addition to its application in IoT, ZKP has been effectively employed in e-health systems. In 2022, Gaba et al. [21] introduced a ZKP-based authenticated key agreement scheme for EHRs, integrating ZKP with biometrics, symmetric cryptography, and message digest techniques.

Despite advancements, existing solutions face key limitations, including reliance on interactive key agreements that hinder scalability, dependence on centralized entities posing security risks, and a lack of formal verifiability, leaving authentication records vulnerable. These challenges underscore the need for a more efficient, decentralized, and verifiable authentication framework.

**Privacy-preserving biometric authentication.** Multimodal biometrics is an emerging technology for biometric authentication. Nowadays, many researchers focus on enhancing its privacy and security [7,22]. Tarannum et al. [23] proposed a multimodal authentication framework that integrates iris, fingerprint, and facial recognition to improve data authentication and security. Later, Zhang et al. [24] implemented the multimodal biometric authentication scheme for Android devices, leveraging facial and voice recognition with an adaptive fusion technique to enhance accuracy. Similarly, Walia et al. [9] developed a framework for generating cancelable multimodal biometric templates, capable of mitigating the impact of low-quality images while preserving image quality attributes.

Despite these advancements, existing approaches face critical challenges in ensuring full-process security protection. Morampudi et al. [7] proposed a privacy-preserving bimodal

authentication system based on homomorphic encryption (HE) to compute the Hamming distance between encrypted biometric templates. However, the adoption of HE introduced significant computational overhead, rendering the authentication system cumbersome. Furthermore, many existing solutions either rely on centralized entities for authentication, introducing single points of failure, or fail to provide formal verifiability, leaving authentication records susceptible to compromise.

To overcome the above weakness, we propose ZABA, an anonymous biometric authentication scheme that integrates MCBG technology with biometric-based ZKP to secure the entire authentication process. Building on our prior work [12], this paper replaces the biometric vector in [12] with an MCBG-based authentication mechanism, further enhancing privacy protection. Unlike existing schemes, ZABA is designed to be computationally efficient while ensuring user anonymity, unlinkability, and resistance against replay attacks. Additionally, it offers forward security, ensuring that past authentication sessions remain uncompromised even if future credentials are exposed. By eliminating reliance on a trusted third party and reducing computational overhead, ZABA provides a scalable and robust solution for privacy-preserving biometric authentication.

## Security model

### System model

Fig 1 shows the simple diagram of our scheme, involving four interactive entities, i.e., the trusted entity (TE), the patient (PA), the authentication and authorization server (AAS), and the doctor (DR). This diagram illustrates the ZABA authentication process, depicting the critical interactions among entities. It highlights how biometric credentials are securely processed and verified to protect PA's anonymity and privacy. The characteristics and roles of each entity are elucidated in this section:

- **TE**: TE is a trusted third party responsible for the initial setup of the entire system. It is liable for generating the public parameters as well as storing the corresponding public key.
- **PA**: The system is installed in the smart phone, and the PA logs in through his fingerprint, face and iris, generates corresponding public and private keys and operates his EHRs.
- **AAS**: AAS is responsible for the authentication of PA seeking access to the AAS. It is responsible for securely storing the PA's biometric data for authentication. The biometric data and EHRs of PAs are saved in ciphertext form to ensure privacy. In our proposed system, the AAS must authenticate the user while maintaining the anonymity of their actual identity.
- **DR**: DR must obtain authorization from PA before viewing their EHRs.

### Threat model

TE serves as a trusted third party in our established threat model. The communication channel connecting TE and the PA is assumed to be secure. Meanwhile, the AAS is characterized as "honest but curious", implying that it adheres to our proposed scheme and responds reliably to queries. However, it may aggressively gather sensitive information, including identity-related data. The channels connecting PA/AAS and PA/DR are deemed insecure, as adversaries could potentially engage in eavesdropping, message interception, replay attacks, and message synthesis. Our model takes into account two kinds of attacks.

- The Insider Attack: User's true identities may be intentionally collected and deduced by DR/AAS administrators or a hacking AAS.

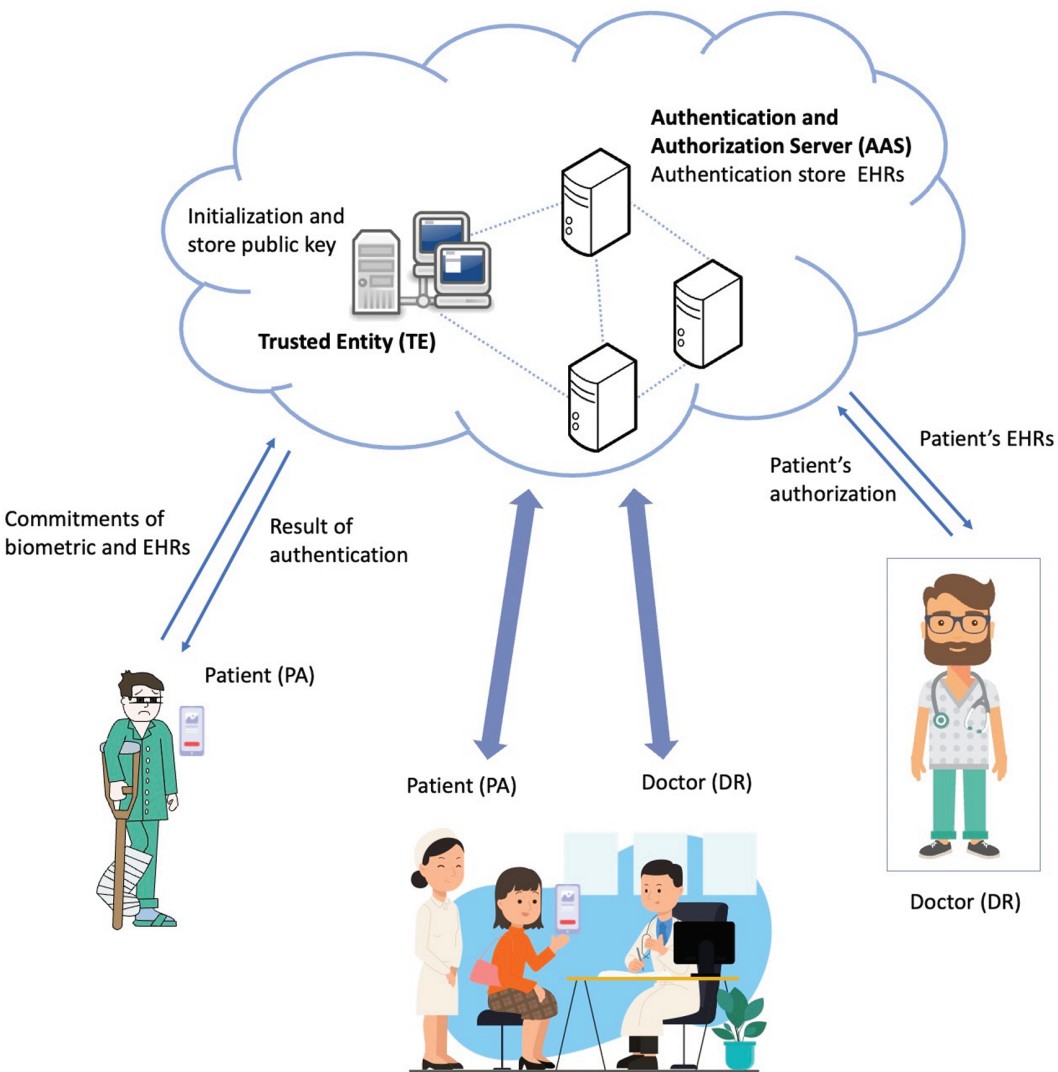

**Fig 1. Simple diagram of the ZABA scheme.**

- The Outsider Attack: The attacker pretends to be a legitimate PA or replays the message to gain access to AAS's services. Additionally, the attacker may observe the message to collect sensitive information about the PA.

We construct a new anonymous authentication scheme to meet the following safety targets to eliminate these potential risks throughout the authentication process in e-health systems.

- Mutual authentication: Our scheme can be used to detect an invalid AAS and an illegal PA.
- Anonymity: The actual identity remains secret during the authentication process among PA and AAS. Potential adversaries may acquire the pseudonyms associated with PAs, but they are unable to deduce any details regarding the actual identity of the PA.
- Unlinkability: Even when AAS fails or controls the wireless channel, adversaries remain unable to identify or establish links to the PA.

- Forward security: Attackers cannot obtain any useful information about the previous session even if they have all the information about the current session.
- Replay attack resistance: Replaying the previous legal messages between the PA and AAS cannot endanger our scheme.

## Preliminaries

Before we state our scheme, we first note some underlying tools.

### Notation

The notations used throughout the paper are listed in Table 1.

### Multimodal cancelable biometric technique

We employ a MCBG technique proposed in [9]. The MCBG utilizes key images derived from cancelable feature generation and adapts them based on their quality to enhance recognition robustness within the adversarial environment. It uses common biometrics for recognition, namely fingerprint ($p$), iris ($i$), face ($f$). Algorithm 1 describes the MCBG technique. It comprises key based generic feature extraction and adaptive weighted graph fusion. Details about the MCBG technique are as follows:

- Key based generic feature extraction: To obtain raw features $\delta^p$, $\delta^f$ and $\delta^i$, input the captured multibiometric data for three modalities. The raw features for key images $K_j^k \forall j \in \{1, 2, \ldots, n\}$, $k \in \{p, f, i\}$ are extracted for three modes of operation: $\psi^p$, $\psi^f$ and $\psi^i$ respectively. Then, association of query images $I^k \in \{I^p, I^f, I^i\}$ with the corresponding key images $K_j^k$ is devoted to generating similarity graphs $G^k \in \{G^p, G^f, G^i\}$. Furthermore, the process of anchored normalization is used in the individual similarity graphs, resulting in the generation of matching normalized graphs, $Q^k \in \{Q^p, Q^f, Q^i\}$.
- Adaptive weighted graph fusion: input the normalised graphs $Q^k$ to construct the sparse graphs $S^k \in \{S^p, S^f, S^i\}$ and the rank graphs $R^k \in \{R^p, R^f, R^i\}$. Using the described nonlinear graph

**Table 1. Notations.**

| Symbol | Representation |
|---|---|
| $[N]$ | The set $\{1, 2, \ldots, N\}$ |
| $p, q$ | The large prime number |
| $n$ | The product of $p, q$ |
| $\mathbb{G}$ | $\mathbb{Z}$ |
| $\mathbb{Z}_n$ | The ring modulo $n$ |
| $\mathbb{G}^j, \mathbb{Z}_n^j$ | The vector spaces of dimension $j$ over $\mathbb{G}$ and $\mathbb{Z}_n$ |
| $\mathbb{Z}_n^*$ | $\mathbb{Z}_n \setminus \{0\}$ |
| $x \xleftarrow{\$} \mathbb{Z}_n^*$ | Sampling uniformly of an element from $\mathbb{Z}_n^*$ |
| $\mathbf{a}$ | $\mathbf{a} \in \mathbb{Z}_n^j$, a vector with elements $a_1, \ldots, a_j \in \mathbb{Z}_n$ |
| $\mathbf{b} = c \cdot \mathbf{a}$ | $b_i = c \cdot a_i$, where $c \in \mathbb{Z}_n$ |
| $\langle \mathbf{a}, \mathbf{b} \rangle$ | $\sum_{i=1}^{j} a_i \cdot b_i$ |
| $\mathbf{P}(x)$ | $\sum_{i=0}^{d} \mathbf{p_i} \cdot x^i \in \mathbb{Z}^j[x]$ |
| $\langle l(x), r(x) \rangle$ | $\sum_{i=0}^{d} \sum_{j=0}^{i} \langle \mathbf{l_i}, \mathbf{r_j} \rangle \cdot x^{i+j} \in \mathbb{Z}[x]$ |
| $\mathbf{a}_{[:\ell]}$ | $(a_1, \ldots, a_\ell) \in \mathbb{Z}_n^\ell, \ell \in [0, s]$ |
| $\mathbf{a}_{[\ell:]}$ | $(a_{\ell+1}, \ldots, a_s) \in \mathbb{Z}_n^{s-\ell}, \ell \in [0, s]$ |
| $C = \mathbf{g}^{\mathbf{a}}$ | $\prod_{i=1}^{j} g_i^{a_i} \in \mathbb{G}$ |

fusion technique, the graphs are distributed to generate fused vectors $\mu^k$ for their respective modalities. The assignment of weights $\lambda^k$ to $\mu^k$ is done adaptively, considering the evaluation of quality metrics for each modality. Finally, output the unified feature vector **w**.

Unlike traditional biometric authentication, which directly stores and matches biometric templates, MCBG [9] applies irreversible transformations to biometric data, preventing attackers from reconstructing or reusing stolen biometric information even if one modality (e.g., fingerprint, face, or iris) is compromised. Within the MCBG framework, key images are utilized to extract generic features for each modality, and these key images can be revoked in case of biometric data compromise, ensuring adaptability and security. Additionally, MCBG employs an adaptive processing approach, where biometric templates dynamically adjust to prioritize high-quality image features while suppressing lower-quality elements, further strengthening authentication reliability and robustness against spoofing attempts.

**Algorithm 1.** MCBG($I^k, K_j^k$) Function

**Input:** $I^k, K_j^k$ for $\forall j \in \{1, 2, \dots, n\}$
**Output:** **w**
1 **for** $k \in \{p, f, i\}$ **do**
2 extract raw features $\delta^k$ for the given value $I^k$;
3 **for** $j \in \{1, 2, \dots, n\}$ **do**
4 extract raw features $\psi_j^k$ for the given value $K_j^k$;
5 **end**
6 **end**
7 **for** $k \in \{p, f, i\}$ **do**
8 **for** $j \in \{1, 2, \dots, n\}$ **do**
9 generate $G_j^k$ using $I^k$ with the corresponding $K_j^k$;
10 normalise $G_j^k$ to $Q_j^k$;
11 **end**
12 generate $S^k$ and $R^k$ from $Q^k$;
13 compute $\mu^k$ and $\lambda^k$ using $S^k$ and $R^k$.
14 **end**
15 compute $\mathbf{w} = <\lambda^p \cdot \mu^p, \lambda^f \cdot \mu^f, \lambda^i \cdot \mu^i>$

## Assumption and commitment

We adopt the Pedersen vector commitment [25] for e-health systems due to its efficiency, privacy, and security advantages. Additionally, it ensures strong hiding and binding properties, preventing unauthorized modifications while supporting privacy-preserving aggregation. To enhance anonymity, we also adopt the Discrete Log (DL) assumption and ZKP to maintain anonymity. Referring to [12,25], we give the definition of DL assumption, RSA group and Pedersen vector commitment in this section.

**Definition 1** (DL Assumption). *For all Probabilistic Polynomial Time (PPT) adversaries $\mathcal{A}$ and $j \geq 2$, there has a negligible function $\mu(\lambda)$ satisfies*

$$\mathrm{P} \left[ \begin{array}{l} \mathbb{G} = \mathrm{Setup}\left(1^\lambda\right), \\ g_1, \dots, g_j \xleftarrow{\$} \mathbb{G}; \\ a_1, \dots, a_j \in \mathbb{Z}_n \leftarrow \mathcal{A}\left(g_1, \dots, g_j\right) \end{array} : \begin{array}{l} \exists a_i \neq 0, \\ \prod_{i=1}^{j} g_i^{a_i} = 1 \end{array} \right] \leqslant \mu(\lambda). \tag{1}$$

As Bünz et al. [25] said, the equation $\prod_{i=1}^{j} g_i^{a_i} = 1$ is a non-trivial DL relation among $g_1, \dots, g_j$. The DL relation assumption ensures that an adversary cannot discover any non-trivial relation among randomly chosen group elements. Notably, this assumption is equivalent to the DL assumption when $j \geq 1$.

**Definition 2** (RSA Group). *In the multiplicative group $\mathbb{G}$ of the integers modulo n, in which n is the product of two large primes p and q. The hardness of computing the order of the group $\mathbb{G}$ is equivalent to the hardness of factoring n.*

**Definition 3** (Commitments). *A non-interactive commitment scheme has two PPT algorithms* (Setup, Com). *The setup algorithm, denoted as $pp \leftarrow \mathrm{Setup}(1^\lambda)$, generates the public parameters pp using the security parameter $\lambda$. The commitment algorithm $\mathrm{Com}_{pp}$ defines* $\mathrm{M}_{pp} \times \mathrm{R}_{pp} \rightarrow \mathrm{C}_{pp}$ *for a message space $\mathrm{M}_{pp}$, a randomness space $\mathrm{R}_{pp}$ and a commitment space $\mathrm{C}_{pp}$ determined by pp. For a message $x \in \mathrm{M}_{pp}$, the algorithm randomly selects $r \xleftarrow{\$} \mathrm{R}_{pp}$, and computes the commitment* $\mathbf{com} = \mathrm{Com}_{pp}(x, r)$.

**Definition 4** (Pedersen Commitment). *$\mathrm{M}_{pp}, \mathrm{R}_{pp} = \mathbb{Z}_n$ and $\mathrm{C}_{pp} = (\mathbb{G}, *)$ being a multiplicative group.*

$$\mathrm{Setup} : g, h \xleftarrow{\$} \mathbb{G}, \tag{2}$$

$$\mathrm{Com}(x; r) = (g^x h^r). \tag{3}$$

**Definition 5** (Pedersen Vector Commitment). *$\mathrm{M}_{pp} = \mathbb{Z}_n^j, \mathrm{R}_{pp} = \mathbb{Z}_n$ and $\mathrm{C}_{pp} = (\mathbb{G}, *)$ being a multiplicative group.*

$$\mathrm{Setup} : \mathbf{g} = (g_1, \dots, g_j), h \xleftarrow{\$} \mathbb{G}, \tag{4}$$

$$\mathrm{Com}(\mathbf{x} = (x_1, \dots, x_j); r) = h^r \mathbf{g}^{\mathbf{x}} = h^r \prod_i g_i^{x_i} \in \mathbb{G}. \tag{5}$$

In the definition above, $r$ is chosen at random. The Pedersen vector commitment for the group $\mathbb{G}$ exhibits perfectly hiding and computationally binding under the DL assumption.

## ZKP-based anonymous biometric authentication scheme

### Structure of the ZABA scheme

Built upon our system model, we construct the ZABA scheme for EHRs. The fundamental objective is to establish authentication between a PA and an AAS while maintaining the anonymity of the PA. The framework of the ZABA scheme appears in Fig 2, encompassing four phases: setup, registration, authentication, and acquisition. In the setup stage, each PA is initialized by TE when the patient initially registers with TE. Subsequently, when PA attempts to access services within the AAS, the authentication process between AAS and PA will be initiated. This authentication process is based on ZKP protocol to realize mutual authentication and hide PA's identity from server administrators. The entire process is described below.

In the ZABA scheme, the security parameter is denoted as $\lambda$, the module $n$ is represented as the product of two large primes, namely $q$ and $p$.

**Setup phase.** This section involves the preparation of the authentication process. TE generates the public parameters $PP = (g, h, \mathbf{g}, \mathbf{h})$ and any participant can ask TE for $PP$. The PA's identity data is only stored in his database, which should be anonymous to AAS. PA/DR's identity data generates his public/private keys and sends the public key to TE. The setup phase is made up of three steps.

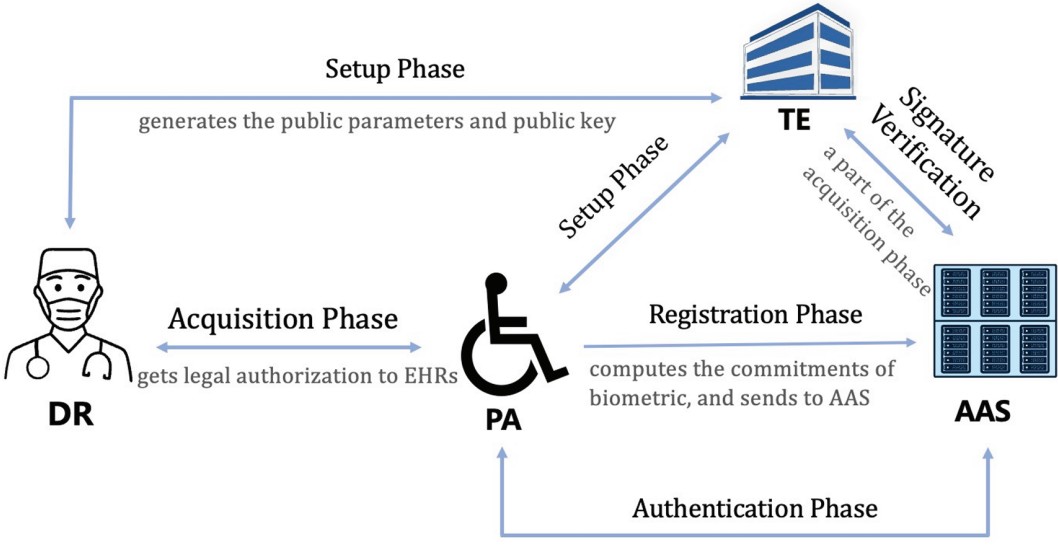

**Fig 2. Overall architectural of the ZABA scheme.**

*Step 1*: Randomly selects $N_p \xleftarrow{\$} \mathbb{Z}_n$, PA computes the private key $sk = H_p(I_{PA}||N_p)$ as well as the public key $pk = g^{sk}$. The identity data for PA is kept as $\{I_{PA}, N_p, sk, pk\}$ in the database. The PA initiates the setup phase by sending the public key pk.

*Step 2*: Upon receiving pk, the TE generates PP=$(g, h, \mathbf{g}, \mathbf{h})$ and stores the tuple $(g, h, \mathbf{g}, \mathbf{h}, pk)$ in the database.

*Step 3*: The DR randomly selects $N_d \xleftarrow{\$} \mathbb{Z}_n$ and computes the private key $sk_d = H_p(I_{DR}||N_d)$ as well as the public key $pk_d = g^{sk_d}$. Then, DR sends $pk_d$ to TE.

**Registration phase.** The registration phase consists of four steps. During these steps, the AAS and the PA acquire secret parameters through the secure channel established between AAS/PA and TE. The specifics of these steps are depicted in Fig 3.

*Step 1*: PA receives the biometric traits, namely fingerprint ($\mathbf{K}^p$), face ($\mathbf{K}^f$), and iris ($\mathbf{K}^i$) from the sensor in phone. Then, PA runs the function MCBG($I^k, \mathbf{K}^k$), $k \in \{p, f, i\}$ to output the feature vector $\mathbf{w}$ and saves it in his database.

*Step 2*: PA randomly selects parameters $\beta \leftarrow \mathbb{Z}_n$, and computes the commitments $W$ of $\mathbf{w}$:

$$W = h^\beta \mathbf{g}^\mathbf{w} \mathbf{h}^\mathbf{w} \in \mathbb{G} \tag{6}$$

*Step 3*: PA signs the tuple $(g, h, \mathbf{g}, \mathbf{h}, W)$ with private key $sk$ and sends it to AAS.

*Step 4*: Upon the receipt of $\sigma(g, h, \mathbf{g}, \mathbf{h}, W)$, AAS requests corresponding public key from the TE. After successful signature verification, AAS saves the tuple $(g, h, \mathbf{g}, \mathbf{h}, W)$ in the database.

**Authentication phase.** To maintain the anonymity of the PA during the authentication phase, we construct a ZKP-based anonymous authentication protocol. Fig 4 illustrates the specifics.

*Step 1*: PA receives new biometric traits from the sensor. Then, PA runs the function MCBG($I'^k, \mathbf{K}'^k$), $k \in \{p, f, i\}$ to output $\mathbf{w}'$. After that, PA computes $\mathbf{d} = \mathbf{w} - \mathbf{w}'$, where $\mathbf{w}$ is stored in the PA's database.

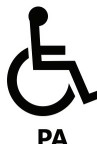

**PA**

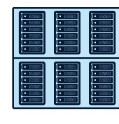

**AAS**

**Step 1:** Inputs the biometric $I^p, I^f, I^i, K^p, K^f, K^i$;

Computes $\mathbf{w} = \text{MCBG}(I^k, K^k), k \in \{p, f, i\}$;

**Step 2:** Choose $\beta \overset{\$}{\leftarrow} \mathbb{Z}_n$;

Computes $W = h^\beta \mathbf{g}^\mathbf{w} \mathbf{h}^\mathbf{w}$;

**Step 3:** Signs $\sigma(g, h, \mathbf{g}, \boldsymbol{h}, W)$ with PA's $sk$;

$$\xrightarrow{\sigma(g, h, \mathbf{g}, \boldsymbol{h}, W)}$$

**Step 4:**
Verifies using PA's public key $pk$

**Fig 3. The step of registration phase.**

*Step 2*: PA randomly generates $\alpha, \theta, \phi, \rho \overset{\$}{\leftarrow} \mathbb{Z}_n$, $\mathbf{s}_L, \mathbf{s}_R \overset{\$}{\leftarrow} \mathbb{Z}_n^m$ and $\mathbf{s}_L', \mathbf{s}_R' \overset{\$}{\leftarrow} \mathbb{Z}_n^6$. After that, PA calculates the following parameter and sends the tuple $(A, D, W', S, S', K)$ to AAS for anonymous authentication:

$$A = h^\alpha \prod_{j=1}^{2} \mathbf{g}_{[3(j-1):3j]}^{j \cdot \mathbf{d}_{[3(j-1):3j]}} \cdot \prod_{j=1}^{2} \mathbf{h}_{[3(j-1):3j]}^{j \cdot \mathbf{d}_{[3(j-1):3j]}} \in \mathbb{G};$$

$$D = h^\alpha \mathbf{g}^\mathbf{d} \mathbf{h}^\mathbf{d} \in \mathbb{G};$$

$$W' = h^\theta \mathbf{g}^{\mathbf{w}'} \mathbf{h}^{\mathbf{w}'} \in \mathbb{G};$$

$$K = h^\phi \in \mathbb{G};$$

$$S = h^\rho \mathbf{g}^{\mathbf{s}_L} \mathbf{h}^{\mathbf{s}_R} \in \mathbb{G};$$

$$S' = h^\rho \mathbf{g}^{\mathbf{s}_L'} \mathbf{h}^{\mathbf{s}_R'} \in \mathbb{G}; \tag{7}$$

*Step 3*: Upon the receipt of $(A, D, W', S, S', K)$, AAS chooses $x, y, z, c \overset{\$}{\leftarrow} \mathbb{Z}_n^*$ and sends to PA.

*Step 4*: PA selects $\tau_1, \tau_2 \overset{\$}{\leftarrow} \mathbb{Z}_n$ and calculates the following parameter:

$$T_i = g^{t_i} h^{\tau_i} \in \mathbb{G}, i \in [2];$$

$$\mathbf{l} = l(x) = \mathbf{d}z - \mathbf{y} + \mathbf{s}_L x \in \mathbb{Z}^m;$$

$$\mathbf{r} = r(x) = \mathbf{d}z + \mathbf{y} + \mathbf{s}_R x \in \mathbb{Z}^m;$$

$$\hat{t} = \langle \mathbf{l}, \mathbf{r} \rangle = t_0 + t_1 x + t_2 x^2 \in \mathbb{Z};$$

$$\tau_x = \tau_2 \cdot x^2 + \tau_1 \cdot x + z^2 r \in \mathbb{Z};$$

$$\mu = \alpha z + \rho x \in \mathbb{Z};$$

$$\mathbf{l}' = z \cdot \sum_{j=1}^{2} j \cdot \left( \mathbf{0}^{3(j-1)} \| \mathbf{d}_{[3(j-1):3j]} \| \mathbf{0}^{3(2-j)} \right) - \mathbf{y} + \mathbf{s}_L' x;$$

**Fig 4. The step of authentication phase.**

$$\mathbf{r}' = z \cdot \sum_{j=1}^{2} j \cdot \left(\mathbf{0}^{3(j-1)} \big\| \mathbf{d}_{[3(j-1):3j]} \big\| \mathbf{0}^{3(2-j)}\right) + \mathbf{y} + \mathbf{s}'_R x;$$

$$\hat{t}' = \langle \mathbf{l}', \mathbf{r}' \rangle \in \mathbb{Z}; \tag{8}$$

in which $t_i$ is $\langle l(x), r(x) \rangle$ corresponding coefficient. Then, PA sends ( $T_1$, $T_2$, $\mathbf{l}$, $\mathbf{r}$, $\hat{t}$, $\tau_x$, $\mu$, $\hat{t}'$, $\mathbf{l}'$, $\mathbf{r}'$) to AAS.

*Step 5*: AAS calculates the following parameter:

$$P = A^z \cdot S^x \cdot \mathbf{g}^{-\mathbf{y}} \cdot \mathbf{h}^{\mathbf{y}} \in \mathbb{G};$$

$$P' = D^z \cdot S^x \cdot \mathbf{g}^{-\mathbf{y}} \cdot \mathbf{h}^{\mathbf{y}} \in \mathbb{G};$$

$$V_1 = V^4 \cdot g^{-4a} \cdot g = g^{4v-4a+1} h^{4r} = g^{y_1} h^{r_1} \in \mathbb{G};$$
$$V_2 = g^{4b} \cdot V^{-4} \cdot g = g^{4b-4v+1} h^{-4r} = g^{y_2} h^{r_2} \in \mathbb{G};$$
$$\mathbf{V} = (V_1, V_2) \in \mathbb{G}^2; \tag{9}$$

Then, AAS outputs "accept" only when the following equations hold:

$$P \stackrel{?}{=} h^{\mu} \cdot \mathbf{g}^{\mathbf{l}} \cdot \mathbf{h}^{\mathbf{r}} \in \mathbb{G};$$
$$P' \stackrel{?}{=} h^{\mu} \cdot \mathbf{g}^{\mathbf{l}'} \cdot \mathbf{h}^{\mathbf{r}'} \in \mathbb{G};$$
$$g^{\hat{t}} h^{\tau_x} \stackrel{?}{=} V^{z^2} g^{-\delta(y)} \cdot T_1^x \cdot T_2^{x^2} \in \mathbb{G};$$
$$\hat{t} \stackrel{?}{=} \langle \mathbf{l}, \mathbf{r} \rangle \in \mathbb{Z};$$
$$\hat{t}' \stackrel{?}{=} \langle \mathbf{l}', \mathbf{r}' \rangle \in \mathbb{Z};$$
$$W^c h^e \stackrel{?}{=} W'^c \cdot D^c \cdot K. \tag{10}$$

**Acquisition phase.** In this phase, DR must obtain authorization from PA before viewing his EHR. There are four steps included in the acquisition phase.

*Step 1*: To sends the request $m$ to the PA, DR signs $m$ with private key $sk_d$ and sends $\sigma(m)$ to AAS.

*Step 2*: Upon receiving $\sigma(m)$ from DR, PA verifies it using DR's public key $pk_d$, which is obtained in advance from TE. After successful verification, PA encrypts his corresponding EHR with DR's public key $pk_d$ and sends $en(\text{EHR})$ to him.

*Step 3*: Upon the receipt of $en(\text{EHR})$, DR decrypts $en(\text{EHR})$ using his private key $sk_d$.

## Security analysis

The security analysis of the ZABA scheme against pertinent potential attacks on the e-health system is provided in this section. Initially, we proved that the ZABA scheme fulfils perfect completeness, perfect special honest verifier zero-knowledge, and computational witness extended emulation, as expounded in [12]. Furthermore, we align our security analysis with the security model in the Security Model section, showing the capability of our proposed system to withstand a range of attacks. These include mutual authentication, user anonymity, forward security, resistance against replay attacks, user unlinkability.

### User anonymity

Preserving user anonymity is vital to our system's design. Even if adversaries gain unauthorized access to the relevant details, particularly the data kept within the AAS, they cannot ascertain the PA's identity. This ensures that the PA's identity remains undisclosed and protected from malicious entities.

**Theorem 1.** *In the ZABA scheme, unauthorized access to the AAS does not enable the attacker to obtain the actual identity of the PA.*

*Proof*: In registration phase, PA generates $W$ according to his own biometric vector $\mathbf{w}$, and sends $(W, g, h, \mathbf{g}, \mathbf{h})$ to AAS. Then, PA generates $W'$ based on the re-extracted biometric vector $\mathbf{w}'$, and sends the $(W', g, h, \mathbf{g}, \mathbf{h})$ to AAS in authentication phase. According to the Pedersen vector commitment and Pedersen commitment, $W$ and $W'$ statistically reveal nothing to AAS. Simultaneously, PA generates proof that the AAS can verify without disclosing the PA's true identity. Even if adversaries intercept the commitments, they cannot infer the PA's real

identity. Additionally, ZABA can effectively mitigate insider attacks, further safeguarding user anonymity.

## Replay attack resistance

Reply attack resistance requires that the ZABA scheme possess the capability to properly detect the replay of prior messages.

**Theorem 2.** *In the ZABA scheme, the attacker can not pass the authentication by replaying the legitimate message from a prior session.*

*Proof* : In our proposed scheme, we have implemented measures to prevent attackers from successfully employing replay attacks to bypass the authentication process. Specifically, if an attacker attempts to replay a previously captured message, denoted as $\sigma(W', g, h, \mathbf{g}, \mathbf{h})$, and they would need to satisfy one of two following conditions.

- $N_p = N_p'$;
- $sk = sk'$, where $sk = H_p(I_{PA}||N_p)$ and $sk' = H_p'(I_{PA}||N_p')$.

Let $N_p$ represent the random parameters by AAS, and let $N_p'$ denote the replayed random parameters selected by AAS. The AAS randomly selects $N_p'$, then $Pr[N_p = N_p']$ holds with a negligible probability. Secondly, the hash function $H_p$ is deemed ideal collision-resistant, which ensures that the attacker cannot produce an identical hash value without access to $N_p$, thus reinforcing the resilience of our scheme against replay attacks.

## Unlinkability

Unlinkability ensures that a user's real identity remains concealed even in the event of a server breach or repeated access to anonymous identities by an attacker.

**Theorem 3.** *In ZABA, attackers can not get the true identity of users from the anonymous authentication messages.*

*Proof* : Pedersen vector commitment is used in the ZABA scheme to protect secret values, notably $W = h^\theta \mathbf{g}^\mathbf{w} \mathbf{h}^\mathbf{w}$, by relying on the computational hardness of the DL assumption. Consider a hypothetical situation where an attacker obtains the value $\mathbf{w}$ from $W$. Nevertheless, the attacker's ability to compute $\mathbf{w}$ from $W$ is hampered by the formidable challenge posed by solving the DL assumption. Furthermore, the user independently chooses random values $\alpha, \beta, \theta, \phi \in \mathbb{Z}_n$ for every iteration within our system. The association between the anonymous identity and the actual user identity remains cryptographically protected, rendering it virtually impossible for the attacker to establish any correlation. Additionally, the ZABA scheme also satisfies the property of zero-knowledge. Hence, the ZABA scheme supports unlinkability.

## Forward security

Forward security guarantees the inability of an attacker to compromise the security of prior authentication procedures, even if they gains access to the current communication information.

**Theorem 4.** *In the ZABA scheme, the attacker is prevented from obtaining information concerning prior sessions through the observation of the current session.*

*Proof*: Forward security guarantees the protection of previous session information in the event of an attacker gaining access to the current session. This is accomplished through the use of unique and non-repetitive random parameters $\{\alpha, \beta, \theta, \phi\}$, which are generated for each session. Additionally, these random parameters ensures that each session possesses a fresh set of secrets, further enhancing the system's resistance against potential attacks.

## Experiments and evaluations

This section provides an overview of our implementation and evaluation, followed by a comparative analysis with other similar anonymous authentication schemes [26–28].

### Implementation

The performance evaluation of the ZABA scheme is conducted on both computer and mobile phone platforms. The simulations are carried out using specific hardware configurations to ensure precision. The principal evaluation device consists of a mobile phone running on the Android 13 operating system with a 3123 MHz CPU and 8 GB RAM. A computer utilizing macOS 11.2.2, powered by Apple M1 and equipped with 16 GB RAM, is also employed for a more comprehensive analysis. To simulate the server environment on both platforms, we employ the C++ library Armadillo and the Java library UJMP.

It is important to note that many factors will influence system performance, and the chosen hardware configurations play a significant role in achieving optimal functionality. Moreover, in order to ensure robust security, the prime numbers $p$ and $q$ are set to 1024 bits. This choice ensures the effectiveness of a Pedersen hash function over an RSA group with $n = p * q$, as per the benchmark. Additionally, the parameter $e$ is set to 7000. For identity authentication, Euclidean distance is chosen as the primary metric due to its effectiveness in measuring similarity between biometric feature vectors while maintaining computational efficiency. Based on our prior analysis [29], the authentication process is considered successful when the Euclidean distance between vectors $\mathbf{w}$ and $\mathbf{w}'$ is less than or equal to 4, ensuring an optimal balance between accuracy and security.

### Test databases

In implementing the ZABA scheme, we utilize benchmark databases to facilitate the process. According to [30], we formulate virtual multibiometric databases for our implementation. Specifically, we use MCYT [31] for the fingerprint, CAS-PEAL [32] for the face, and IITD PolyU [33] for the iris. Meanwhile, distinct and individualized mapping is maintained across unimodal databases to simplify the creation of a multimodal database. To ensure the validity of the ZABA scheme, it is essential that the same number of subjects be used for fingerprint, face, and iris. In our discussions, a unique one-to-one mapping among the first $N$ subjects across different databases is maintained to generate the multimodal database. Meanwhile, this information is converted into a vector of a certain length on the local side, facilitating further analysis and processing.

For the dataset of EHRs, we employ MIMIC-III [34], which is a publicly accessible database and widely used in healthcare research. To enhance the realism of our simulations, we also establish a one-to-one correspondence between the multibiometric database above and the EHRs dataset, thereby closely mimicking actual healthcare environments. However, real-time data acquisition poses a significant challenge due to variations in data collection

frequency and potential delays in updating patient records. To address this, future work will explore adaptive data synchronization techniques and real-time processing optimizations to enhance the system's responsiveness and applicability in dynamic healthcare settings.

## Experimental evaluation and results

In this section, we delve into the performance evaluation of the ZABA scheme, a solution that enables anonymous authentication in e-health systems. The presented data represents the average result derived from conducting 10,000 experiments. The ZABA scheme's communication cost remains consistent, indicated by $11\mathbb{G} + 7\mathbb{Z}_n + (2m + 19)\mathbb{Z}$, as confirmed by our computational analysis.

Fig 5 presents line charts that illustrate the proving time and verification time of the ZABA scheme, respectively. This chart demonstrates the system's running time across varying Euclidean distances. The results indicate that both proving and verification times remain stable as the Euclidean distance increases, with an average proving time of approximately 78 ms and a verification time of around 140 ms. This consistency suggests that the proving and verification processes are largely unaffected by the range of Euclidean distances, ensuring the scheme's efficiency and scalability.

Furthermore, we evaluated our scheme more specifically within a healthcare environment using a subset of the MIMIC-III dataset [34]. As MIMIC-III is fully de-identified and multimodal, it maintains rigorous participant anonymity and closely mirrors real-world hospital data conditions. In this setting, we measured both the time cost (proving and verification time) and communication cost per patient under data sizes typical of clinical workflows, as illustrated in Fig 6. Our results confirm that the ZABA scheme delivers acceptable running times and network usage levels for practical deployment, meeting throughput and latency

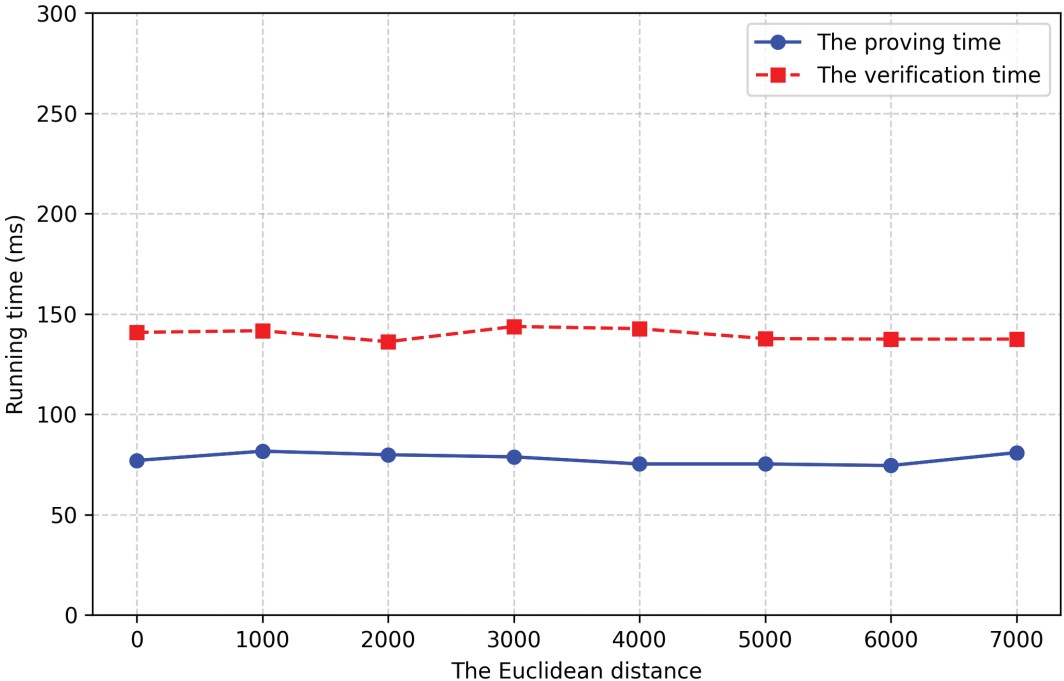

**Fig 5. The running time.**

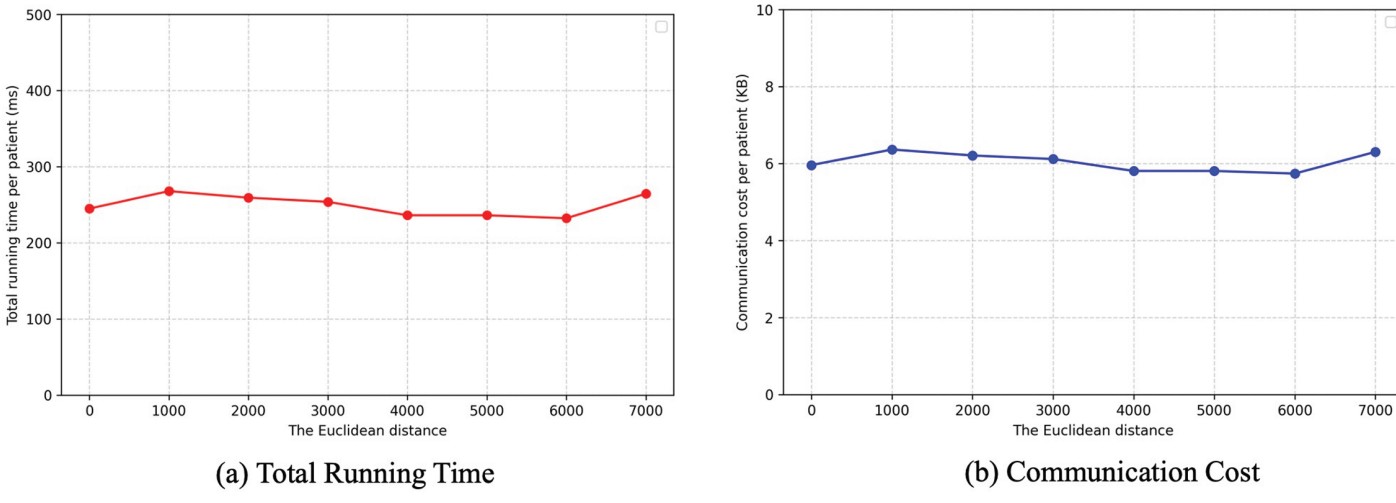

(a) Total Running Time                                        (b) Communication Cost

**Fig 6. Performance evaluation in a healthcare environment.**

demands while preserving patient privacy. These results demonstrate the scheme's readiness for secure, real-time data analysis in clinical decision-making scenarios.

### Comparative analysis with existing schemes

For the purpose of evaluating the ZABA scheme, we conduct a comparative analysis between our scheme and existing representative authentication schemes, as illustrated in Table 2. We consider three comparison criteria: security, assumption and time complexity. In Table 2, the symbol "✓" signifies that the corresponding property is fulfilled, while "✗" indicates that the property is not met.

In Table 2, the result shows that our scheme is more secure. Boussada's [28], Yang's [26], Aghili's [27], and ours can provide anonymity for e-health systems. Maintaining patient's anonymity is of utmost importance in e-health systems. But Yang's [26] and Aghili's [27] schemes fail to provide unlinkability. And Yang's [26] scheme fails to provide forward security. In contrast, only our scheme uses multimodal biometrics to establish a more comprehensive and reliable identification system. Regarding security assumptions, while the security of the aforementioned schemes [26–28] rely on the DL assumption, our scheme is based on both the DL assumption and the RSA group, as elaborated in Definitions 1 and 2. Our scheme's security is slightly better than that of the others.

**Table 2. Comparison results of similar schemes.**

| Scheme | Security analysis | | | Comparison | | |
|---|---|---|---|---|---|---|
| | Anonymity | Replay attack resistance | Unlinkability | Forward security | Biometric authentication | Assumption |
| Boussada's [28] | ✓ | ✓ | ✓ | ✓ | ✗ | DL |
| Yang's [26] | ✓ | ✓ | ✗ | ✗ | ✗ | DL |
| Aghili's [27] | ✓ | ✓ | ✗ | ✓ | ✗ | DL |
| Our scheme | ✓ | ✓ | ✓ | ✓ | ✓ | RSA+DL |

## Conclusion

To address the challenges of patient privacy and authentication in e-health, we propose a patient-centered, anonymous e-health system in this paper. We use the ZKP protocol to enable anonymous biometric verification, ensuring that patient identities remain confidential during authentication. Additionally, we utilize multimodal biometrics to enhance the overall security and reliability of the verification process. Combining these technologies, we aim to create a robust and patient-centric anonymous biometric authentication scheme for e-health systems. The evaluation simulates real-world scenarios in a healthcare environment, showing ZABA's potential for secure and efficient authentication in e-health systems.

To further enhance the robustness and applicability of our proposed scheme, future research will focus on addressing certain limitations identified in this study. The use of a single hardware configuration constrains the scope of our evaluation, and conducting comparative analyses across diverse hardware platforms would provide a more comprehensive assessment of system performance. Additionally, the dataset employed in our experiments is relatively small compared to the volumes processed in daily applications. These may affect the generalizability of our research. To mitigate these concerns, future work will explore testing on larger datasets and optimizing the scheme for low-power devices to enhance its efficiency and feasibility in resource-constrained environments, such as IoT-based healthcare systems.

## Author contributions

**Conceptualization:** Ying Chen.

**Data curation:** Xiaqing Zhou.

**Methodology:** Xuechun Mao.

**Resources:** Xuechun Mao.

**Writing – original draft:** Xuechun Mao.

**Writing – review & editing:** Xuechun Mao, Xiaoming Zhao, Ying Chen.

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
