## [Decision Letter · Decision Letter 0]

19 Feb 2025

PONE-D-24-56772A ZKP-Based Anonymous Biometric Authentication Scheme

for the E-Health SystemsPLOS ONE

Dear Dr. Chen,

Thank you for submitting your manuscript to PLOS ONE. After careful consideration, we feel that it has merit but does not fully meet PLOS ONE’s publication criteria as it currently stands. Therefore, we invite you to submit a revised version of the manuscript that addresses the points raised during the review process.

We look forward to receiving your revised manuscript.

Kind regards,

Veer Singh, Ph.D

Academic Editor

PLOS ONE

Journal Requirements:

6. Please amend either the abstract on the online submission form (via Edit Submission) or the abstract in the manuscript so that they are identical.

**Additional Editor Comments:**

Dear Author,

Your manuscript is not yet complete for publication. Therefore, I suggest a major revision of your manuscript. Please respond carefully to each of the reviewers' comments.

Reviewers' comments:

Reviewer's Responses to Questions

**Comments to the Author**

1. Is the manuscript technically sound, and do the data support the conclusions?

Reviewer #1: Yes

Reviewer #2: Partly

2. Has the statistical analysis been performed appropriately and rigorously? 

Reviewer #1: Yes

Reviewer #2: No

3. Have the authors made all data underlying the findings in their manuscript fully available?

Reviewer #1: Yes

Reviewer #2: Yes

4. Is the manuscript presented in an intelligible fashion and written in standard English?

Reviewer #1: Yes

Reviewer #2: Yes

5. Review Comments to the Author

Reviewer #1: Please add the contribution of all the authors and acknowledge the funding agency. The authors have written the manuscript in a well-mannered manner with substantial data and discussion. The data of the current manuscript is enough for publication in journal like PLOS one

Reviewer #2: The manuscript titled ‘A ZKP-Based Anonymous Biometric Authentication Scheme for E-Health Systems’ is well-structured and presents an interesting approach. However, several aspects require significant improvement. I suggest revisions that will enhance its clarity and impact.

• The abstract should be revised to remove redundant descriptions of the benefits of e-health systems, like improved accessibility, efficiency, and patient experience and the security aspects of the proposed scheme. Additionally, the phrase ‘and more’ is vague and unnecessary, and should be omitted for clarity.

• I suggest highlighting specific contributions or key insights gained from this study in the abstract. This will help differentiate your study from existing literature reviews on the topic and emphasize its novelty.

• Define each acronym like ‘IoT’ at its first use. Check through the entire manuscript to make sure it is defined at the first use.

• The manuscript discusses multimodal biometrics and cancelable biometrics (MCBG) for security, given that multimodal biometrics can still be vulnerable to presentation attacks. However, it does not mention any anti-spoofing techniques. How does your scheme handle biometric spoofing attacks?

• The English of the manuscript needs to be improved.

• The conclusion section should be revised and concise.

6. PLOS authors have the option to publish the peer review history of their article (what does this mean?). If published, this will include your full peer review and any attached files.

Reviewer #1: No

Reviewer #2: No

---

## [Author Response · Author response to Decision Letter 1]

18 Mar 2025

Responds to the reviewer's comments:

Reviewer #1:

Please add the contribution of all the authors and acknowledge the funding agency. The authors have written the manuscript in a well-mannered manner with substantial data and discussion. The data of the current manuscript is enough for publication in journal like PLOS one.

Response: Thank you for your positive feedback and suggestion. We have included the contribution details of all authors and acknowledged the funding agency in the Cover Letter. We appreciate your recognition of our work and the quality of our data and discussion.

1. Page 1, Lines 8-15 Comment: The abstract should provide specific details about the methodology, such as key metrics (e.g., accuracy, computational cost) and evaluation outcomes. Replace vague phrases like "the effectiveness and practicality of our scheme" with concrete results.

Response: Thank you for your suggestion. We have revised the abstract by clearly specifying the key metrics to present concrete results. To enhance clarity and precision, we have replaced vague expressions with explicit descriptions that better reflect our scheme.

2. Page 1, Line 20, and throughout the manuscript Comment: Ensure consistent use of the acronym "AAS" for "Authentication and Authorization Server" throughout the manuscript. Double-check for any remaining instances of "ASS."

Response: Thank you for your suggestion. We have thoroughly reviewed and revised the manuscript to ensure the consistent use of the acronym "AAS" for "Authentication and Authorization Server."

3. Page 2, Lines 10-15 Comment: Expand on how the proposed ZABA scheme improves upon prior ZKP-based solutions, particularly in terms of efficiency, security, and practical deployment in e-health systems.

Response: Thank you for your suggestion. As your recommendation, we have highlighted how the proposed ZABA scheme improves upon previous ZKP-based solutions in the contributions section.

4. Page 3, Lines 5-10 Comment: Add a summary that explicitly identifies gaps in the literature and connects them directly to the proposed contributions of this study.

Response: Thank you for your suggestion. We have revised the Related Work section to explicitly identify gaps in the literature and establish a direct connection between these gaps and the contributions of our study.

5. Figures (Pages 9 and 12) Comment: Ensure all figure captions are placed below the figures and provide self-explanatory details. For example, the "Simple Diagram of the ZABA scheme" caption should briefly explain its significance in the authentication process.

Response: Thank you for your suggestion. We have carefully revised the figure captions to ensure they are correctly positioned below the corresponding figures. Additionally, we have provided self-explanatory details for all figures.

6. Page 8, Lines 15-20 Comment: Provide a clearer justification for choosing the Pedersen vector commitment algorithm over other cryptographic methods. Discuss its specific advantages in the context of e-health.

Response: Thank you for your suggestion. In the Assumption and Commitment section, we provide a reason for selecting the Pedersen vector commitment algorithm.

7. Page 14, Line 5 Comment: Explain why Euclidean distance is chosen as the primary metric for identity authentication and how the threshold (≤4) was determined.

Response: Thank you for your suggestion. We have added an explanation in the Implementation section, clarifying the rationale behind our choice.

8. Throughout the manuscript (e.g., Page 2, Line 5) Comment: Revise grammar and syntax issues, such as "The e-health system oers" to "The e-health system offers." Perform a thorough language edit to improve readability.

Response: Thank you for your suggestion. We have carefully revised the manuscript to correct grammar and syntax issues.

9. Page 15, Lines 10-15 Comment: Discuss the challenges faced during experimental implementation, such as computational overhead or real-time data acquisition limitations, and their potential resolutions.

Response: Thank you for your suggestion. We have expanded the discussion in the Test Databases section to address the challenges encountered during experimental implementation.

10. Page 16, Lines 5-10 Comment: Expand the conclusion by briefly discussing future enhancements, such as testing with larger datasets or optimizing the scheme for low-power devices.

Response: Thank you for your suggestion. In response, we have expanded the conclusion to discuss potential future enhancements.

Special thanks to you for your good comments.

Reviewer #2:

The manuscript titled ‘A ZKP-Based Anonymous Biometric Authentication Scheme for E-Health Systems’ is well-structured and presents an interesting approach. However, several aspects require significant improvement. I suggest revisions that will enhance its clarity and impact.

1. The abstract should be revised to remove redundant descriptions of the benefits of e-health systems, like improved accessibility, efficiency, and patient experience and the security aspects of the proposed scheme. Additionally, the phrase ‘and more’ is vague and unnecessary and should be omitted for clarity.

Response: Thank you for your suggestion. We have revised the abstract to remove redundant descriptions and vague terms, ensuring clarity.

2. I suggest highlighting specific contributions or key insights gained from this study in the abstract. This will help differentiate your study from existing literature reviews on the topic and emphasize its novelty.

Response: Thank you for your suggestion. We have revised the abstract to highlight the key contributions and novel insights of our study.

3. Define each acronym like ‘IoT’ at its first use. Check through the entire manuscript to make sure it is defined at the first use.

Response: Thank you for your suggestion. We have carefully reviewed the manuscript and ensured that each acronym is defined upon its first use for clarity and consistency.

4. The manuscript discusses multimodal biometrics and cancelable biometrics (MCBG) for security, given that multimodal biometrics can still be vulnerable to presentation attacks. However, it does not mention any anti-spoofing techniques. How does your scheme handle biometric spoofing attacks?

Response: Thank you for your suggestion. We have included a detailed explanation in the Multimodal Cancelable Biometric Technique section. Specifically, we emphasize the use of irreversible transformations, key image revocation, and adaptive processing techniques to enhance resilience against presentation attacks.

5. The theoretical analysis section should focus on evaluation rather than theoretical analysis. This evaluation can also involve testing the approach in a healthcare environment and analyzing the results.

Response: Thank you for your suggestion. We have added a detailed performance evaluation in the Experimental Evaluation and Results section. Specifically, we have incorporated an assessment of our scheme in a healthcare environment using the MIMIC-III dataset.

6. The English of the manuscript needs to be improved.

Response: Thank you for the suggestion. We have carefully revised the manuscript to enhance clarity, refine grammar, and improve overall readability.

7. The conclusion section should be revised and concise.

Response: Thank you for your suggestion. We have revised the conclusion section to enhance clarity and conciseness.

Special thanks to you for your good comments.

We appreciate for Editors/Reviewers' warm work earnestly and hope that the correction will meet with approval.

Once again, thank you very much for your comments and suggestions.

---

## [Decision Letter · Decision Letter 1]

23 Apr 2025

A ZKP-Based Anonymous Biometric Authentication Scheme

for the E-Health Systems

PONE-D-24-56772R1

Dear Dr. Ying Chen

We’re pleased to inform you that your manuscript has been judged scientifically suitable for publication and will be formally accepted for publication once it meets all outstanding technical requirements.

Kind regards,

Veer Singh, Ph.D

Academic Editor

PLOS ONE

Reviewers' comments:

Reviewer's Responses to Questions

**Comments to the Author**

1. If the authors have adequately addressed your comments raised in a previous round of review and you feel that this manuscript is now acceptable for publication, you may indicate that here to bypass the “Comments to the Author” section, enter your conflict of interest statement in the “Confidential to Editor” section, and submit your "Accept" recommendation.

Reviewer #1: All comments have been addressed

Reviewer #2: All comments have been addressed

2. Is the manuscript technically sound, and do the data support the conclusions?

Reviewer #1: Yes

Reviewer #2: Yes

3. Has the statistical analysis been performed appropriately and rigorously? 

Reviewer #1: Yes

Reviewer #2: Yes

4. Have the authors made all data underlying the findings in their manuscript fully available?

Reviewer #1: Yes

Reviewer #2: Yes

5. Is the manuscript presented in an intelligible fashion and written in standard English?

Reviewer #1: Yes

Reviewer #2: Yes

6. Review Comments to the Author

Reviewer #1: All the comments have been carefully addressed. I think the manuscript is ready for publication in its current form. Authors have addressed the comment, but in the response sheets they only included the thanks for the review. I also request the authors place all the comments in reviewer sheet for easy and convenient for the reviewer.

Reviewer #2: Overall, the manuscript is scientifically robust and addresses a critical gap in the healthcare system. With minor revisions to enhance data presentation and clarify statistical methodologies, the paper has the potential to make a substantial contribution to the field.

7. PLOS authors have the option to publish the peer review history of their article (what does this mean?). If published, this will include your full peer review and any attached files.

Reviewer #1: No

Reviewer #2: No

---

## [Editor Report · Acceptance letter]

PONE-D-24-56772R1

PLOS ONE

Dear Dr. Chen,

I'm pleased to inform you that your manuscript has been deemed suitable for publication in PLOS ONE. Congratulations! Your manuscript is now being handed over to our production team.

Kind regards,

on behalf of

Dr. Veer Singh

Academic Editor

PLOS ONE